# Rethinking LoRA Aggregation for Federated Fine-tuning of Foundation Models

## Abstract

The application of Low-Rank Adaptation (LoRA) in Federated Learning (FL) systems provides an effective solution for Foundation Models (FMs) to leverage distributed private data. However, the heterogeneous distribution of client-side data has hindered the performance of federated systems from reaching. Through an in-depth investigation of this issue, we discover that LoRA parameter aggregation among clients gives rise to fine-grained conflicts and introduces the cross-term noise interference for subsequent rounds. Both factors disadvantage the efficient convergence of federated fine-tuning performance. Based on these findings, we propose a **H**armonious **F**ederated **Lo**w-**R**ank **A**daption method (HFLoRA), which first detects conflicts in LoRA row update directions between clients through a fine-grained joint regulation mechanism, then imposes inhibitory constraints on anomalous conflict rows using scaling factors. This effectively resolves the LoRA aggregation conflict issue in FL and provides a theoretical proof. In addition, we have designed a global LoRA consistent re-decomposition strategy that further mitigates the impact of cross-term noise on FL by computing a pair of optimal low-rank matrices from the aggregated noise-free global LoRA. HFLoRA is also applicable to federated environments with heterogeneous LoRA and does not introduce additional communication costs. Extensive experiments across natural language generation and vision tasks demonstrate that HFLoRA consistently outperforms other state-of-the-art FL methods on different benchmarks. Our code is available at: https://anonymous.4open.science/r/HFLoRA .

## 1 Introduction

Federated Learning (FL) ( McMahan et al. (2017); Yang et al. (2024b); Ye et al. (2023); Kim et al. (2024)), as a distributed machine learning paradigm, has emerged as a key technology for collaborative model training on privacy-sensitive data. In recent years, researchers have increasingly focused on methods that integrate federated learning with Foundation Models (FMs) training (Fan et al. (2023); Kuang et al. (2024); Wu et al. (2024a)), given its potential to effectively address the challenges posed by the growing volume of training data and the impending depletion of publicly available data. However, deploying large-scale pre-trained models with billions of parameters (such as LLaMA (Touvron et al. (2023)) and GPT-4 (Achiam et al. (2023))) within a federated learning framework presents significant challenges, including the substantial fine-tuning training overhead required by clients and the enormous communication costs involved.

Parameter Efficient Fine-Tuning (PEFT) (Ding et al. (2023); Xu et al. (2023)) technology has been introduced into federated learning to significantly reduce computational resources during FMs fine-tuning. Among these, Low-Rank Adaptive (LoRA) (Hu et al. (2022)) is highly favored for its outstanding performance and simplicity. LoRA indirectly updates model weights by optimizing a set of low-rank matrices, thereby reducing the number of parameters that need to be updated by several orders of magnitude. Although LoRA significantly enhances computational and communication efficiency in FL (Wu et al. (2024b); Damle et al. (2025); Han et al. (2024); Luo et al. (2024); Wang et al. (2025)), there remain two fundamental issues when applying LoRA for FL fine-tuning FMs: *(a) LoRA row*[1] ***aggregation conflicts between clients:*** As shown in Fig.1a, we observe that under heterogeneous data conditions, the row $a \in \Delta W$ update direction conflict values of the LoRA

---

[1]The LoRA row $(a)$ represent the row parameters of the LoRA weight matrix $\Delta W$ (Eq.2).

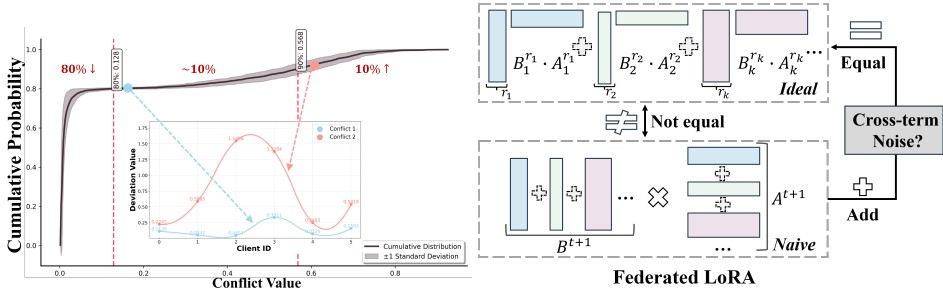

(a) Cumulative Probability of Conflict      (b) Cross-term Noise after Aggregation

Figure 1: Illustration of (a) The cumulative probability distribution of conflict difference values among different rows $a$ of the LoRA weight matrix $\Delta W = BA$, where only a few rows $< 10\%$ exhibit significant optimization conflicts. The inset visualizes detailed information about two specific conflict points; (b) During LoRA aggregation on the FL server, unknown cross-term noise exists between naive aggregation and ideal aggregation that is unknown to all clients.

weight matrix $\Delta W = BA$ obtained through client-side optimization exhibit a long-tail distribution pattern ($80\% < 0.12$, while $10\% > 0.56$). Among these, some updates to the LoRA row parameters may point to local extrema that contradict the global optimum, and can be regarded as "conflict noise." If the LoRA low-rank matrices are weighted averaged indiscriminately, this effectively injects such noise into the global LoRA (Kuo et al. (2024)). ***(b) Presence of cross-term noise after LoRA aggregation:*** As shown in Fig.1b, there exists an unknown cross-term noise between the results of the naive and the ideal aggregation for LoRA between clients, which causes clients to deviate from the optimal global consensus starting point when initializing LoRA parameters in a new round. This will affect the stability and convergence speed of subsequent training. Furthermore, when the rank ($r_1 \neq r_2$) of LoRA differs among clients, many existing FL aggregation algorithms also prove difficult to deploy (Cho et al. (2024)).

Some studies (Sun et al. (2024); Fan et al. (2025); Bian et al. (2024)) have noted this issue this issue. Most recently, Yan et al. (2025) proposes FRLoRA, which communicates weight changes $\triangle W'$ during each round and updates the local model accordingly. However, this approach significantly increases communication costs ($Params(\triangle W') \gg Params(B + A)$), consuming more communication resources. Furthermore, these methods have not thoroughly analyzed the structural relationships of LoRA parameters among clients during federated aggregation, making it challenging to effectively address the issue of client LoRA parameter drift under data heterogeneity. Therefore, there is an urgent need for a fine-grained aggregation method specifically designed for federated LoRA to better facilitate the collaborative training of large-scale language models.

**Contributions.** In this paper, our research focuses on exploring how to efficiently aggregate client LoRA parameters in the federated fine-tuning FMs. Under the heterogeneous distribution, the fine-tuned parameters across different clients exhibit multiple gradient directions (Zheng et al. (2025)). In section 3, we conduct a fine-grained analysis of the structural differences in LoRA among clients. First, we analyze the variability in the update directions of LoRA parameters, identifying the row conflict issue ***(a)***. Subsequently, regarding the new round of reallocation for LoRA, we mathematically calculate the difference between naive aggregation and ideal aggregation methods, revealing the nature of cross-term noise ***(b)***. In FL, heterogeneous data distribution further amplifies this noise.

To address this, we propose a **H**armonious **F**ederated **Lo**w-**R**ank **A**daption method (**HFLoRA**), which is applicable to heterogeneous LoRA without incurring additional communication costs and harmoniously resolves the aforementioned issues. ***First***, HFLoRA conducts row conflict detection on the update directions of LoRA weight matrices $\Delta W$ among clients through the Fine-grained Joint Regulation. It employs a designed scaling factor $\gamma$ to suppress the anomalous conflicting rows, thereby effectively addressing issue ***(a)***. Theoretically, we prove the effectiveness and convergence of this regulation. ***Second***, during the aggregation process, the Global LoRA Consistent Re-decomposition strategy performs noise-free aggregation on the regulated LoRA weight matrix $\Delta \widetilde{W}$, rather than aggregating the low-rank matrices $A$ and $B$, thereby avoiding interference from cross-term noise ***(b)***. The subsequent re-decomposition algorithm ensures that clients ultimately

obtain the optimal initial LoRA low-rank matrix parameters $(A^{t+1}, B^{t+1})$ for the next round, guaranteeing the consistency and continuity of the LoRA model parameter space.

We summarize our key contributions as follows: (**1**) We investigate the parameter conflicts across clients and cross-term noise issues in LoRA aggregation within federated learning, offering insights into the causes of performance degradation. (**2**) We propose a novel method, HFLoRA, which mitigates LoRA row aggregation conflicts and implements the global LoRA consistent re-decomposition to enhance FL performance under data heterogeneity. (**3**) Extensive experiments on natural language understanding and vision tasks demonstrate the superiority of HFLoRA over existing methods.

## 2 RELATED WORKS

### 2.1 FEDERATED LEARNING

Federated Learning (FL) (Li et al. (2020b); Wang et al. (2024a); Li et al. (2021b); Yang et al. (2024b); Kim et al. (2024)), as a distributed machine learning paradigm, aims to collaboratively train a global model across multiple clients while ensuring data privacy. Due to its effective protection of user data privacy, FL has found widespread applications in fields such as healthcare and finance, where data security is paramount (Seo et al. (2024); Li et al. (2020a); Nguyen et al. (2022); Antunes et al. (2022); Kumar & Singla (2021); Li et al. (2021a)). However, data heterogeneity, characterized by the non-independent and identically distributed nature of local data, poses a significant challenge for the practical deployment of FL (Ye et al. (2023); Qu et al. (2022); Mendieta et al. (2022); Luo et al. (2021)). To address this challenge, the research community has proposed numerous strategies, including the regularization terms (Li et al. (2020b)); the adoption of personalization (Li et al. (2021c)); and the robust weight aggregation (Ma et al. (2022b)). While these methods have achieved notable success in FL, most studies neglect the unique parameter structures associated with parameter-efficient fine-tuning methods such as LoRA. Moreover, directly integrating LoRA with FL for fine-tuning FMs has proven difficult in achieving optimal performance (Ye et al. (2024a)).

### 2.2 FEDERATED FINE-TUNING WITH LORA

Low-Rank Adaptation (LoRA) (Hu et al. (2022); Hayou et al. (2024); Agiza et al. (2024); Yang et al. (2024a)) is a parameter efficient fine-tuning technique that has emerged as a critical research focus in the domain of FL, particularly for fine-tuning FMs due to its significant reduction in the number of parameters involved. Due to LoRA's updating mechanism, the combination of LoRA with federated learning for fine-tuning Foundation Models significantly reduces the demand for local computational resources while achieving strong performance (Ding et al. (2024)). By transmitting small parameter matrices, communication costs are significantly reduced. Recent years have seen initial successes in applying LoRA for fine-tuning FMs within FL contexts (Bian et al. (2024); Sun et al. (2024); Wang et al. (2024b); Yan et al. (2025); Guo et al. (2025a); Fan et al. (2023)). Initially, FedBERT (Tian et al. (2022)) utilizes FL and split learning to train the pre-trained BERT model. Zhang et al. (2022) explores benchmarks for PEFT in the fine-tuning of FMs in FL. Zhang et al. (2024) conducts the first investigation into FL instruction tuning tasks tailored, introducing FedIT. This method performs separate aggregation only on low-rank matrices $(B, A)$ during aggregation, which introduces cross-term noise issues for Clients. OpenFedLLM (Ye et al. (2024b)) establishes a multi-FL algorithm library for fine-tuning large language models. However, these approaches primarily aggregate low-rank matrices, which has led to the emergence of the cross-term noise issues. To address this, Sun et al. (2024) proposes FFA-LoRA, which enhances training stability by fine-tuning only the zero matrix $A$. However, this has somewhat diminished LoRA's learning capabilities. Wang et al. (2024b) introduces FLoRA, achieving adaptability through a stacked approach for heterogeneous LoRA, but this inherently increases communication costs. Guo et al. (2025a) put forward FedSA-LoRA, enabling client-specific training by only transmitting matrix $B$. FRLoRA (Yan et al. (2025)) mitigates client drift by sharing the residual parameters of the LoRA weight matrix. Most of these methods remain applicable solely within homogeneous rank settings or incur substantial communication overhead, and there is still a lack of in-depth exploration concerning the aggregation of LoRA's unique structure. Therefore, addressing the issues of LoRA parameter aggregation fine-grained conflicts and the cross-term noise within federated fine-tuning FMs remains an urgent challenge that needs to be resolved.

## 3 LoRA in Federated Learning

In this paper, our research focuses on exploring how to more effectively aggregate the LoRA parameters fine-tuned on clients. The following will conduct a granular analysis of issues present in the LoRA fine-tuning FMs between federated clients.

**Preliminary.** Assume there is a federated system comprising $K$ clients and a secure central server, where each client $k$ possesses a private dataset $D_k$. The clients collaborate to fine-tune pre-trained model $W_0 \in \mathbb{R}^{d \times l}$, with each client deploying trainable LoRA low-rank matrix ($B_k \in \mathbb{R}^{d \times r}, A_k \in \mathbb{R}^{r \times l}, r \ll min(d,l)$). And the naive federated fine-tuning FMs training structure is as follows:

$$
\begin{aligned}
Clients: \quad & W_0 + \Delta W_k = W_0 + B_k^t A_k^t, \quad \forall k \in K \\
Server: \quad & B^{t+1} = \sum_{k=0}^{K} p_k B_k^t, \quad A^{t+1} = \sum_{k=0}^{K} p_k A_k^t \,,
\end{aligned}
\tag{1}
$$

where $p_k$ is the aggregation weight, and $B^{t+1}$ and $A^{t+1}$ indicate the new round of parameters after aggregation. The optimization objective for the client is given by $f_k = \mathcal{L}(D_k; W_0 + B_k A_k)$, where $\mathcal{L}$ is the empirical loss function. Therefore, the global optimization objective of FL is expressed as $\mathcal{F} = \frac{1}{K} \sum_{k=1}^{K} f_k$. However, this approach in Eq.1 presents two fundamental issues.

***Defect (a): LoRA Row Aggregation Conflicts Between Clients.*** Due to the heterogeneity of data in FL, servers encounter fine-grained row conflicts when aggregating LoRA parameters. Previous studies (Lin et al. (2025)) have shown that a significant challenge when fine-tuning LoRA models for downstream tasks is the issue of overfitting. In FL, this overfitting problem persists due to the differences in data distribution, leading to discrepancies in the direction of LoRA updates.

In the forward propagation process, the row parameter $a \in \mathbb{R}^{1 \times l}$ of the LoRA weight matrix $\Delta W$ can be viewed as independent low-dimensional projection transformations of the original high-dimensional feature space (He et al. (2025)). To explore this, we transform the client's LoRA weight matrix $\Delta W_k$ into row parameter $a$ form to study the LoRA structure in FL:

$$
\Delta W_k \in \mathbb{R}^{d \times l} = B_k \in \mathbb{R}^{d \times r} \cdot A_k \in \mathbb{R}^{r \times l} = [a_k^1, .., a_k^j, .., a_k^d]^T,
\tag{2}
$$

where $\Delta W_k$ denotes the LoRA parameter matrix for client $k$, and $a_j \in \mathbb{R}^{1 \times l}$ represents the LoRA row parameters. Thus, the transformation logic of the LoRA model for data $x^t$ is as follows: $x^{t+1} = (W_0 + \Delta W_k)x^t = W_0 x^t + [a_k^1 x^t, .., a_k^j x^t, .., a_k^d x^t]^T$. Different rows are independent of each other.

Inspired by this, we perform conflict value detection for update direction in LoRA row parameters between clients (Section 4.1). The results are illustrated in Fig.1a, which presents the cumulative distribution of LoRA row parameter conflict detection during a single random federated learning communication process. The vertical axis represents the cumulative probability distribution of the row parameters, while the horizontal axis represents the conflict values of row update directions among different clients. The results show that approximately $10\% - 20\%$ of the LoRA row parameters exhibit significant direction conflicts (red curve), while about $80\%$ of the updates achieve consensus in update direction among different clients (blue curve).

This indicates that when different clients perform local updates on LoRA parameters, there are significant conflicts in the optimization directions of a minority of rows. Directly aggregating these conflicting anomalous parameters would negatively impact the performance of federated learning (Guo et al. (2025b)). Therefore, it is essential to introduce a fine-grained regulatory mechanism to guide the optimization of LoRA parameters among different clients toward a consensus direction.

***Defect (b) : Presence of Cross-term Noise after LoRA Aggregation.*** In federated fine-tuning, noise-free aggregation of clients' LoRA parameters poses a significant challenge. The commonly used naive aggregation approach, which aggregates matrices $B$ and $A$ separately, introduces an unknown cross-term noise to the clients. This is primarily due to the fact that the backpropagation loss is calculated based on the LoRA weight matrix $\Delta W_k$. Efficiently aggregating low-rank matrices without incurring additional communication costs presents substantial difficulties.

In this, we calculate this aggregation using mathematical formulas. Considering the heterogeneity of data in FL, the comparison between the results of naive aggregation of client LoRA parameters

and the ideal aggregation results is presented as follows:

$$\underbrace{W_0 + (p_1 B_1 A_1 + ... + p_K B_K A_K)}_{Ideal\ aggregation} \neq \underbrace{W_0 + (p_1 B_1 + ... + p_K B_K)(p_1 A_1 + ... + p_K A_K)}_{Naive\ aggregation}, \quad (3)$$

where $p_k$ is the parameter aggregation weight and $\sum_{k=0}^{K} p_k = 1$. The difference between the left and right sides of Fq.3 indicates that naive aggregation introduces cross-term noise into the updated parameters. Next, we present the results of the cross-term noise calculation (In *Appendix* B):

$$Ideal - Naive = \sum_{k=1}^{K} p_k (B_k - \bar{B})(A_k - \bar{A}), \ where \ \bar{A} = \sum_{k=0}^{K} p_k A_k, \ \bar{B} = \sum_{k=0}^{K} p_k B_k. \quad (4)$$

In Eq.4, the cross-term noise will only disappear if the low-rank matrices $B$ and $A$ of all clients remain consistent before and after aggregation. However, such a situation does not exist in the context of federated learning. Furthermore, the heterogeneity of client data further exacerbates this issue. Mitigating cross-term noise becomes a critical problem that needs to be addressed.

# 4 PROPOSED METHOD: HFLoRA

In the previous section, we discuss two main issues regarding LoRA in federated aggregation. Inspired by this, we propose the HFLoRA method, which enhances federated collaboration performance through the Conflict Fine-grained Joint Regulation and the Global LoRA Consistent Redecomposition strategy.

## 4.1 CONFLICT FINE-GRAINED JOINT REGULATION

The LoRA aggregation conflict issue for **Defect (a)** may lead to a slowdown in the overall convergence speed of the federated system. To address this issue, we propose a Conflict Fine-grained Joint Regulation mechanism for the LoRA parameters. Specifically, after the server receives the LoRA parameters during each communication round, it first computes the parameter weight matrix for each client as $\Delta W_k = B_k \cdot A_k = [a_k^1, a_k^2, ..., a_k^j, ..., a_k^d]^T$. Consequently, we obtain the joint set of all client parameters regarding the $j$-th row of LORA:

$$\mathcal{A}_j = \{a_1^j, ..., a_k^j, ..., a_K^j\}, \ where \ |\mathcal{A}_j| = K, \quad (5)$$

where $a_k^j$ denotes the LoRA $j$-th row parameter of client $k$. To resolve conflicts among different LoRA rows $\{\mathcal{A}_j\}$ between clients, we need to make the following common consensus update assumption 4.1 in distributed optimization algorithms.

**Assumption 4.1 (Consensus Update).** *Assume there exists a potential global optimal row update direction $\mathbf{a}_*^j$, and local updates $a_k^j$ from clients can be expressed as:*

$$a_k^j = \mathbf{a}_*^j + \mathbf{n}_k^j + \mathbf{s}_k^j; \quad \mathbf{n}_k^j \sim \mathcal{N}(0, \sigma_k^2 I_d), \quad (6)$$

*where $\mathbf{n}_k^j$ is the instability of the optimization and $\mathbf{s}_k^j$ is a systematic bias due to the skewness in client data distribution. The expected norm of the systematic bias should be small, i.e., $\mathbb{E}[|\mathbf{s}_k^j|2] \ll |\mathbf{a}_*^j|^2$ , ensuring that the update directions $a_k^j$ from clients are concentrated around $\mathbf{a}_*^j$. If updates are dominated by noise and bias, the observed values $\mathcal{C}_j$ will show a high conflict value.*

For the computation of the observation value $\mathcal{C}_j$, we select the commonly used cosine distance as the metric. To focus more on the differences in parameter directions and eliminate the effects of magnitude, we apply $l_2$ normalization to each row vector, expressed as $\hat{\mathbf{a}}_k^j = \frac{a_k^j}{|a_k^j|_2}$. To accelerate the computation of $\mathcal{C}$, we calculate the central position vector $\boldsymbol{\mu}_j = \frac{1}{K} \sum_{k=1}^{K} \hat{\mathbf{a}}_k^j$ of the normalized row parameters from different clients as an anchor point for the average direction. This approach reduces the computational complexity of the average cosine distance from $\mathcal{O}(K^2)$ to $\mathcal{O}(K)$.

Based on this anchor point, we can obtain the conflict observation value $\mathcal{C}_j$ for the update direction by calculating the expected cosine distance between the row parameters $\mathcal{A}_j$ of different clients and the anchor point $\boldsymbol{\mu}_j$. And the conflict value for the $j$-th row is given by:

$$\mathcal{C}_j = \mathbb{E}_{\hat{\mathbf{a}}_k^j \in l_2(\mathcal{A}_j)} \left[ 1 - \hat{\mathbf{a}}_k^j \cdot \boldsymbol{\mu}_j^T \right] . \quad (7)$$

In Eq.7, a larger value of $\mathcal{C}_j$ indicates greater divergence in the optimization direction of the $j$-th row parameters among different clients. The cumulative distribution of the conflict values is illustrated in Fig.1a, where a few rows ($\leq 20\%$) exhibit extreme conflict issues that impact the performance of federated LoRA aggregation. To address this, we introduce a statistical method to identify the anomalous conflicting rows:

$$\begin{cases} \mathcal{H}^{\uparrow} \ni j & , \forall \, \mathcal{C}_j > \delta \\ \mathcal{H}_{\downarrow} & , other \end{cases} \tag{8}$$

where $\mathcal{H}^{\uparrow}$ denotes the set of anomalous conflicting rows; $\mathcal{H}_{\downarrow}$ represents the optimized consensus row; and $\delta$ serves as a specific threshold. Influenced by the experimental results (Fig.1a), we set $\delta$ as the percentile $85\%$ of the statistical distribution of the conflict values $\mathcal{C}$. This threshold will be adjusted in accordance with the changes in parameter distribution during each round of aggregation, accommodating the dynamic instability and convergence of the model training at different stages.

To mitigate the issue of anomalous conflicting rows $\mathcal{H}^{\uparrow}$ arising from noise and bias during training in Consensus Update Eq.6, we introduce a scaling factor $\gamma$ for joint regulation of LoRA rows to suppress the problem of conflicting rows parameter aggregation. We employ an exponential decay function to construct this scaling factor $\gamma_j$:

$$\gamma_j = exp(-\lambda \cdot (\mathcal{C}_j - \delta)) \cdot (1 - I_j) + 1 \cdot I_j \,, \tag{9}$$

where $I_j = \mathbb{I}(\mathcal{H})$ is the indicator function; $\lambda$ is the hyperparameter for decay intensity. When the conflict ($j \in \mathcal{H}^{\uparrow}, I_j = 0$), updates to the anomalous conflicting rows are suppressed ($\gamma_i \to 0$) to achieve approximate sparsity; during consensus updates ($j \in \mathcal{H}_{\downarrow}, I_j = 1$), the row parameters are retained ($\gamma_i = 1$) in their entirety. After scaling, the server obtains the client $k$'s regulated LoRA matrix $\Delta \widetilde{W}_k = \Delta W_k \cdot [\gamma_1, ..., \gamma_j, ..., \gamma_d]^T$, resolving aggregation conflicts.

**Theorem 4.1 (Variance Reduction).** *The proposed HFLoRA method tightens the upper bound of the expected variance of the global update by effectively suppressing the contribution from conflicting rows.* (Proof details in *Appendix* A.1.)

**Theorem 4.2 (Convergence).** *After $T$ rounds of communication, the proposed algorithm converges to a steady state and satisfies:*

$$\frac{1}{T} \sum_{t=1}^{T} \mathbb{E}[||\nabla \mathcal{F}(W_t)||^2] \leq \mathcal{O}(\frac{1}{\sqrt{KT}}) + \mathcal{O}(\frac{\sigma_l^2 + \gamma \cdot \sigma_g^2}{K}) + \mathcal{O}(\xi). \tag{10}$$

Detailed derivations are provided in *Appendix* A.2. Theorem 4.2 provides the convergence rate of HFLoRA, which matches the best convergence rate of existing FL methods. It should be noted that our regulation mechanism shares certain similarities with sparse regularization for single LoRA (Liang et al. (2025)). Our innovation lies primarily in exploring the multiple client LoRA aggregation conflicts inherent in federated learning, enabling more effective solutions to specific challenges.

## 4.2 GLOBAL LoRA CONSISTENT RE-DECOMPOSITION

To address the ***Defect (b)*** of cross-term noise in the aggregation of client LoRA low-rank matrices, we propose a global LoRA consistent re-decomposition strategy. Specifically, we can aggregate the regulated LoRA weight matrices to obtain a noise-free global LoRA $\Delta W_{global} = \sum_{k=1}^{K} p_k \Delta \widetilde{W}_k$. This aggregation process is applicable to weight $p_k$ aggregation methods in other federated learning studies ( Li et al. (2023); Ma et al. (2022a); Fang & Ye (2022)). Subsequently, the consistent re-decomposition process of the global LoRA can be formalized as an optimization problem:

$$\min_{B^{t+1}, A^{t+1}} ||B^{t+1} \cdot A^{t+1} - \Delta W_{global}||_F^2 + \tau(||A^{t+1} - \sum_k^K p_k A_k^t|| + ||B^{t+1} - \sum_k^K p_k B_k^t||), \tag{11}$$

where $|| \cdot ||_F$ denotes the Frobenius norm. The first term of Eq.11 ensures that the product of the new low-rank matrices $B^{t+1}$ and $A^{t+1}$ after federated aggregation can optimally approximate the global LoRA parameters $\Delta W_{global}$, thereby further mitigating the cross-term noise present in LoRA aggregation and ensuring nearly lossless transfer of global knowledge; The second term acts as a consistent regularization to prevent the optimized $B^{t+1}$ and $A^{t+1}$ from deviating from the original parameters. Unlike the approximate solving methods (LoRA-FAIR Bian et al. (2024), FlexLoRA

Table 1: Performance of the proposed method with other state-of-the-art methods on the Alpaca-GPT4 and OpenR1-Math datasets for natural language generation tasks using pre-training LLAMA model. Best results are highlighted in bold.

| Method | Alpaca-GPT4 | | | | OpenR1-Math | | |
|---|---|---|---|---|---|---|---|
| | MMLU | Vicuna | MT-Bench-1 | MT-Bench-2 | Math500 | Gpqa | GSM8K |
| Local | 38.5±1.41 | 5.13±1.36 | 3.77±1.16 | 1.93±0.59 | 36.4±2.82 | 24.3±1.51 | 33.67±3.13 |
| FedIT | 42.3±0.97 | 5.97±0.58 | 4.12±0.74 | 2.45±0.45 | 47.4±1.24 | 27.7±1.62 | 36.65±2.36 |
| LoRA-FAIR | 43.8±1.24 | 6.41±0.67 | 4.84±0.93 | 2.53±0.36 | 51.5±1.58 | 29.7±1.46 | 38.82±1.72 |
| FFA-LoRA | 42.7±1.05 | 6.29±0.78 | 4.72±0.62 | 2.39±0.43 | 51.9±2.38 | 28.8±1.77 | 37.24±2.53 |
| FlexLoRA | 43.6±2.46 | 6.57±1.42 | 4.57±0.81 | 2.49±0.33 | 53.1±1.84 | 29.1±2.17 | 39.32±1.27 |
| FLoRA | 43.1±0.92 | **6.73**±0.85 | 4.86±0.48 | 2.72±0.28 | 52.7±2.73 | 31.4±2.78 | 41.54±0.96 |
| FedSA-LoRA | 42.1±2.16 | 6.37±1.19 | 4.57±0.62 | 2.58±0.48 | 50.5±2.52 | 30.6±1.39 | 39.18±1.73 |
| FRLoRA | 43.5±1.48 | 6.62±0.72 | 4.97±0.46 | 2.67±0.24 | 52.4±2.08 | 31.2±1.62 | 41.59±2.63 |
| **Ours** | **45.9**±1.26 | 6.67±0.47 | **5.21**±0.52 | **3.18**±0.37 | **55.3**±1.71 | **32.3**±1.16 | **42.67**±1.29 |

Bai et al. (2025)), this term additionally ensures that the low-rank matrix across different rounds remains within the same parameter space, thereby facilitating a smoother training process.

Notably, our HFLoRA does not require the same rank $r$ for clients' LoRA, making it applicable to the aggregation problem with varying LoRA parameter ranks ($r_1 \neq r_2$) in FL, as shown in Tab.4. Additionally, our approach does not involve transmitting extra parameters (Tab.3), keeping communication costs aligned with traditional methods while theoretically offering better aggregation decomposition.

## 5 EXPERIMENTS

In this section, we evaluate and compare the performance of the proposed method with other approaches across two types of tasks: natural language generation (NLG) and visual tasks (VT). For natural language generation, we utilize the pre-training LLAMA model (Touvron et al. (2023)) for fine-tuning and conduct experiments on the Alpaca-GPT4[2] (Peng et al. (2023)) ($\approx 52k\ samples$) and OpenR1-Math datasets[3] ($\approx 94k\ samples$). In the case of visual tasks, we employ a pre-trained Vision Transformer (ViT) model (Wu et al. (2020)) to perform scene recognition across 365 categories using the subset of Places365 dataset (Zhou et al. (2017)) ($\approx 1.8M\ samples$). All experiments are conducted using half precision to enhance efficiency. More details in *Appendix* C.

**Baselines.** We compare our method against 7 state-of-the-art baselines, including **FedIT** (Zhang et al. (2024)), **LoRA-FAIR**((Bian et al. (2024)), **FFA-LoRA**(Sun et al. (2024)), **FLexLoRA**(Bai et al. (2025)), **FLoRA**(Wang et al. (2024b)), **FedSA-LoRA**(Guo et al. (2025a)),and **FRLoRA**(Yan et al. (2025)). **Local**: clients train independently without participating in federated learning.

### 5.1 NATURAL LANGUAGE GENERATION

**Experimental Setup.** We utilize the pre-trained LLAMA 2 (7B parameters) from the huggingFace transformers library as our base model in the NLG task. The Alpaca-GPT4 dataset is employed for training and evaluated on multi-task benchmarks, including MMLU(Hendrycks et al. (2021)), Vicuna Bench(Chiang et al. (2023)), and MT-Bench(Zheng et al. (2023)). For the open-ended assessments, we employ GPT (Achiam et al. (2023)) for evaluation. The OpenR1-Math dataset is tested using the Lighteval library (Habib et al. (2023)) on mathematical benchmark datasets, specifically Math500 (Lightman et al. (2023)), Gpqa (Rein et al. (2024)), and GSM8K (Cobbe et al. (2021)). The datasets are randomly partitioned into 10 and 20 clients according to a Dirichlet distribution ($\beta$). Each client utilizes the AdamW optimizer with an initial learning rate $\eta$ set at 5e-5, which is reduced to 1e-6 in the final round. The rank for LoRA is set to 16, with the scalar $\alpha$ set to 32. We maintain a batch size of 16, with local update steps set to 10 and a total of 200 communication rounds across all experiments.

---

[2]https://huggingface.co/datasets/vicgalle/alpaca-gpt4
[3]https://github.com/huggingface/open-r1

Table 2: Performance of the proposed method with others on the Places365 dataset for visual tasks using the pre-training ViT model. $K$ indicates the number of clients; Smaller $\beta$ values reflect greater data heterogeneity.

| Method | $K = 10$ | | | | $K = 30$ | | | |
|---|---|---|---|---|---|---|---|---|
| | $\beta = 0.5$ | | $\beta = 1$ | | $\beta = 0.5$ | | $\beta = 1$ | |
| | Top-1 acc. | Top-5 acc. | Top-1 acc. | Top-5 acc. | Top-1 acc. | Top-5 acc. | Top-1 acc. | Top-5 acc. |
| Local | 39.92±2.32 | 70.50±3.42 | 40.11±2.58 | 70.67±3.89 | 40.05±2.46 | 70.77±4.31 | 40.13±3.43 | 70.83±4.23 |
| FedIT | 46.62±1.32 | 73.90±2.62 | 47.86±1.76 | 74.15±2.58 | 47.15±2.29 | 73.05±3.95 | 47.73±1.25 | 74.86±2.59 |
| LoRA-FAIR | 48.46±2.52 | 76.59±3.74 | 49.59±1.26 | 78.22±2.14 | 49.22±1.68 | 77.42±3.46 | 49.70±1.62 | 79.43±2.57 |
| FFA-LoRA | 49.63±1.83 | 77.34±2.71 | 50.55±1.53 | 82.35±2.47 | 51.20±1.56 | 81.92±2.77 | 51.69±1.47 | 82.18±2.33 |
| FlexLoRA | 49.31±1.36 | 79.49±2.63 | 50.18±1.31 | 82.28±2.16 | 50.99±1.72 | 82.97±2.17 | 51.81±1.94 | 83.62±2.62 |
| FLoRA | 50.03±0.93 | 79.51±1.61 | 51.23±0.84 | 82.04±1.73 | 52.04±1.82 | 83.25±2.36 | 52.39±1.37 | 83.63±1.83 |
| FedSA-LoRA | 49.86±1.13 | 76.98±2.29 | 50.90±1.45 | 82.82±2.29 | 51.63±1.26 | 83.19±1.95 | 52.34±1.83 | 83.72±1.53 |
| FRLoRA | 50.56±0.87 | 81.38±1.79 | 51.27±0.94 | 82.99±1.76 | 52.22±1.58 | 83.36±1.93 | 52.96±1.27 | 83.98±1.93 |
| **Ours** | **52.24**±0.58 | **83.32**±1.45 | **52.37**±1.19 | **83.54**±1.92 | **54.21**±1.46 | **84.73**±2.27 | **54.71**±1.16 | **85.32**±1.68 |

Table 3: Comparison of parameter upload ↑ and download ↓ quantities for client in the LLaMA model ($\approx 7B$) and support for heterogeneous ranks LoRA in FL settings ($K = 20, r = 16$).

| Method | FedIT | LoRA-FAIR | FFA-LoRA | FlexLoRA | FLoRA | FedSA-LoRA | FRLoRA | Ours |
|---|---|---|---|---|---|---|---|---|
| Params(M)-$Up$ ↑ | 8.3M | 8.3M | 4.2M | 8.3M | 8.3M | 4.2M | 8.3M | 8.3M(0.12%) |
| Params(M)-$Down$ ↓ | 8.3M | 8.3M | 4.2M | 8.3M | 167.8M | 4.2M | 536.9M | 8.3M |
| Hetero-ranks $r$ | ✗ | ✗ | ✗ | ✓ | ✓ | ✗ | ✓ | ✓ |

**Results.** As shown in Tab.1, we present the federated results using the pre-trained LLAMA 7B model under the heterogeneous partitioning of the Alpaca-GPT4 and OpenR1-Math datasets. Under these conditions, the LoRA parameter drift among clients is more pronounced. The FedIT method performs the worst due to its inability to address conflict and cross-term noise issues. In contrast, HFLoRA effectively addresses this issue, thereby significantly improving performance across all benchmarks. For example, our method achieves an average improvement of approximately $8\%$ on the Math500 benchmark. FLoRA and FRLoRA are the most competitive methods, but their communication costs are significantly higher than our method (In Tab.3). Overall, our method demonstrates a clear advantage compared to other methods, confirming its effectiveness in addressing natural language generation task.

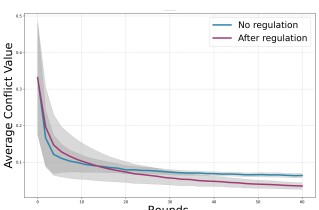

Figure 2: Visualization of the average conflict value across all layers of regulated LoRA as the number of rounds changes.

## 5.2 VISUAL TASKS

**Experimental Setup.** We utilize a pre-trained ViT (86.6M parameters) as the base model (Wu et al. (2020)). Following previous research (Liu et al.), we partition the training set of the Places dataset according to the Dirichlet distribution ($\beta$). The evaluation metric involves testing the global model on the Places365 test dataset, reporting Top-1 and Top-5 accuracy. To accommodate the 365-class scene recognition task, we add two linear layers to the pre-trained ViT model. The learning rates $\eta$ for LoRA and the classification head are set to 1e-4 and 1e-2, respectively. The rank $r$ of LoRA is set to 16, with a scalar $\alpha$ of 32. The local steps is 1, and the batch size is 96.

**Results.** As shown in Tab.2, we conduct federated learning experiments using the ViT model on the Places365 dataset with varying degrees of data heterogeneity $\beta$ and numbers $K$ of clients. Under two distinct types of data heterogeneity in FL, our method generally outperforms other state-of-the-art approaches. As data heterogeneity shifts from $\beta = 1$ to $\beta = 0.5$, most methods exhibit a significant decline in performance compared to our method HFLoRA. This is primarily due to the divergence in local LoRA update directions caused by data heterogeneity, leading to fine-grained conflicts during the aggregation of LoRA parameters. Our approach is specifically designed to address this issue. Additionally, the experiments with different numbers of clients ($K = 10, 30$) are based on subsets of the Places365 dataset ($\approx$0.3M,0.9M $samples$) for federated heterogeneous partitioning. The addition of more clients improves the overall performance of federated learning, demonstrating the advantages of conducting federated learning with a large number of clients. In

Table 4: Performance comparison of different heterogeneous ranks ($r$) LoRA settings and other adaptation methods on the Places365 dataset ($K$=15).

| Hetero-ranks | FedIT | FlexLoRA | FLoRA | FRLoRA | Ours |
|---|---|---|---|---|---|
| $r \in \{4, 8, 16\}$ | 44.82(73.9) | 50.94(82.7) | 51.65(83.2) | 51.96(83.6) | **53.16**(83.9) |
| $r \in \{16, 32, 64\}$ | 45.15(74.7) | 51.26(83.1) | 51.82(83.5) | 51.87(83.4) | **53.42**(84.2) |

Table 5: Results of ablation study on Alpaca-GPT4, OpenR1-Math, and Places365 datasets.

| Method | Alpaca-GPT4 | | | OpenR1-Math | | | Places365 | |
|---|---|---|---|---|---|---|---|---|
| | MMLU | Vicuna | MT-Avg | Math500 | Gpqa | GSM8K | Top-1 acc. | Top-5 acc. |
| $\omega/o\ \mathcal{V}_1$ | 43.52 | 6.52 | 3.95 | 51.74 | 31.16 | 40.53 | 51.84 | 82.69 |
| $\omega/o\ \mathcal{V}_2$ | 44.36 | 6.48 | 4.01 | 53.68 | 31.28 | 41.26 | 52.78 | 83.35 |
| **HFLoRA** | **45.94** | **6.67** | **4.19** | **55.32** | **32.36** | **42.67** | **54.71** | **85.32** |

the configuration with more clients, our method demonstrates higher performance, confirming its robustness and feasibility for deployment in large-scale client environments.

### 5.3 In-Depth Analyses

**Comparison of Methods.** In Tab.3, we compare the communication costs for a single client upload and download, as well as the adaptability to heterogeneous settings, across all baseline methods. For methods with homogeneous rank, our HFLoRA has a communication cost comparable to that of other methods, with computational costs similar to those of LoRA-FAIR. Among methods adapted for heterogeneous ranks, our approach (8.3M) needs lower communication parameter costs compared to other methods, such as FLoRA (167.8M) and FRLoRA (536.9M).

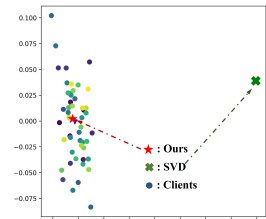

Figure 3: Visualization of low-rank matrix $B$ decomposition comparison.

**LoRA Heterogeneous Ranks $r$.** In Tab.4, we conduct federated experiments with combinations of different heterogeneous ranks. Due to its inability to adapt to heterogeneous rank setting, the FedIT method exhibits extremely poor performance. Overall, our approach outperforms other heterogeneous LoRA methods, further validating its effectiveness in addressing LoRA aggregation challenges under heterogeneous conditions. Simultaneously, we observe that higher ranks enhance federated fine-tuning performance but may induce overfitting.

**Ablation Study.** We conduct ablation experiments on two designs: $\omega/o\ \mathcal{V}_1$ does not employ the Conflict Fine-grained Joint Regulation, while $\omega/o\ \mathcal{V}_2$ replaces the Global LoRA Consistent Re-decomposition with singular value decomposition (SVD). The experimental results are presented in Tab.5. The performance of $\mathcal{V}_1$ is generally low, indicating that the joint regulation mechanism of HFLoRA effectively mitigates the impact of data heterogeneity. In Fig.2, we visualize the average conflict values of LoRA for each round, showing that our regulation mechanism can reduce the upper limit of conflicts in each round. The performance decline of $\mathcal{V}_2$ suggests that our LoRA re-decomposition strategy effectively addresses the instability of SVD initialization in each round, thereby facilitating a smoother federated learning training process. In Fig.3, the parameter of matrix $B$ is compared after decomposition. Our strategy maintains consistency in the parameter space before and after updates by the regularization term. More in *Appendix* C.1.

## 6 Conclusion

In this work, we thoroughly explore the LoRA row conflict phenomenon and cross-term noise issues associated with client LoRA parameter aggregation in federated fine-tuning FMs. To address these challenges, we propose a new Harmonious Federated Low-Rank Adaptation method, named HFLoRA, which resolves LoRA row conflicts by suppressing anomalous rows through the Conflict Fine-grained Joint Regulation. Subsequently, HFLoRA employs a Global LoRA Consistent Re-decomposition strategy to initialize low-rank matrix parameters for each round, facilitating smoother training. Extensive experimental results demonstrate that HFLoRA outperforms other state-of-the-art methods across various LoRA configurations.

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

# A  ADDITIONAL THEORETICAL ANALYSIS

We view the FL LoRA training process as a distributed optimization problem seeking the global optimum $W$ on a non-convex function. The global objective function is:

$$\min_W F(W) = \frac{1}{K} \sum_{k=1}^{K} E_{x \sim D_k}[\ell(W; x)], \tag{12}$$

where, $K$ denotes the number of clients, and $D_k$ represents the local data distribution of the $k$th client.

To facilitate the theoretical analysis, we introduce the following standard and reasonable assumptions:

Assumption 1 (L-smoothness): The local loss function $\ell(W; x)$ for each client is L-smooth, meaning its gradient is Lipschitz continuous:

$$\|\nabla\ell(W; x) - \nabla\ell(W'; x)\| \le L\|W - W'\|, \quad \forall W, W', x. \tag{13}$$

Assumption 2 (Bounded Gradient): At any point $W$, the second-order moment of the stochastic gradient is bounded:

$$\mathbb{E}\left[\|\nabla\ell(W; x)\|^2\right] \le G^2, \quad \forall W, x. \tag{14}$$

Assumption 3 (Bounded Variance): The random variance between local client gradients and the global gradient is bounded. For any client $k$, there exists a constant $\sigma_l^2$ such that:

$$\mathbb{E}_{x \sim D_k}\left[\|\nabla\ell(W; x) - \nabla F_k(W)\|^2\right] \le \sigma_l^2, \quad \forall k, W. \tag{15}$$

Simultaneously, the gradient variance between different clients (reflecting data heterogeneity) is also bounded, with a constant $\sigma_g^2$ such that:

$$\frac{1}{K} \sum_{k=1}^{K} \|\nabla F_k(W) - \nabla F(W)\|^2 \le \sigma_g^2, \quad \forall W. \tag{16}$$

Assumption 4 (Separability of Conflicting Updates): There exists a quantifiable difference $\xi > 0$ such that the expected deviation of harmful updates (i.e., row directions identified as anomalous by

our policy) arising from non-iid data and conflicting with the global optimal direction is at most $\xi$. That is, for a labeled row $j$, we have:

$$\mathbb{E}\left[\|\mathbf{a}_k^j - \mathbf{a}_*^j\|\right] \leq \xi, \tag{17}$$

where, $\mathbf{a}_i^*$ represents the globally consensus-updated vector for that row.

### A.1 PROOF OF THEOREM 4.1

Let $\mathbf{g}^{(k)} = \text{vec}(\Delta W_k)$ denote the incremental update vector for the LoRA parameters of client $k$. The standard LoRA aggregation is:

$$\bar{\mathbf{g}} = \frac{1}{K}\sum_{k=1}^{K}\mathbf{g}^{(k)}. \tag{18}$$

Our strategy can be viewed as applying a diagonal scaling matrix $\Gamma^{(k)}$ (whose diagonal elements consist of scaling factors $\gamma_i$) to each client update, followed by aggregation:

$$\tilde{\mathbf{g}} = \frac{1}{K}\sum_{k=1}^{K}\Gamma^{(k)}\mathbf{g}^{(k)}. \tag{19}$$

According to Assumption 3, we have:

$$\mathbb{E}\|\bar{\mathbf{g}} - \mathbb{E}[\bar{\mathbf{g}}]\|^2 = \mathbb{E}\left\|\frac{1}{K}\sum_k(\mathbf{g}^{(k)} - \mathbb{E}[\mathbf{g}^{(k)}])\right\|^2 = \frac{1}{K^2}\sum_k\mathbb{E}\left\|\mathbf{g}^{(k)} - \mathbb{E}[\mathbf{g}^{(k)}]\right\|^2 \leq \frac{1}{K}\left(\sigma_l^2 + \sigma_g^2\right). \tag{20}$$

For our strategy, the scaling matrix $\Gamma^{(k)}$ is designed such that for rows with high consensus update, $\gamma_j = 1$; for conflicting rows, $\gamma_j \ll 1$. Let $\mathcal{C}$ denote the set of all indices pointing in the direction of conflicting rows. Then the aggregated variance is:

$$\mathbb{E}\|\tilde{\mathbf{g}} - \mathbb{E}[\tilde{\mathbf{g}}]\|^2 = \mathbb{E}\left\|\frac{1}{K}\sum_k\Gamma^{(k)}(\mathbf{g}^{(k)} - \mathbb{E}[\mathbf{g}^{(k)}])\right\|^2 \leq \frac{1}{K}\left(\sigma_l^2 + \gamma \cdot \sigma_g^2\right). \tag{21}$$

By comparing Eq.21 and Eq.20, our approach reduces the variance in global update expectations by effectively suppressing conflicting rows.

### A.2 PROOF OF THEOREM 4.2

We start from Assumption 1 of L-smoothness. This assumption states that the gradient of the function $F$ is Lipschitz continuous, which directly implies that the following inequality holds for all $W_t$ and $W_{t+1}$:

$$F(W_{t+1}) \leq F(W_t) + \langle\nabla F(W_t), (W_{t+1} - W_t)\rangle + \frac{L}{2}\|W_{t+1} - W_t\|^2. \tag{22}$$

According to the algorithm's update rule, we have $W_{t+1} = W_t - \eta\tilde{\mathbf{g}}_t$. Substituting this into the above equation:

$$F(W_{t+1}) \leq F(W_t) + \langle\nabla F(W_t), (W_{t+1} - W_t)\rangle + \frac{L}{2}\|W_{t+1} - W_t\|^2$$

$$= F(W_t) + \langle\nabla F(W_t), (-\eta\tilde{\mathbf{g}}_t)\rangle + \frac{L}{2}\|-\eta\tilde{g}_t\|^2 \tag{23}$$

$$= F(W_t) - \eta\langle\nabla F(W_t), \tilde{\mathbf{g}}_t\rangle + \frac{L\eta^2}{2}\|\tilde{g}_t\|^2.$$

Next, we take the conditional expectation of all randomness in the $t$th round on both sides of the inequality:

$$\mathbb{E}_t[F(W_{t+1})] \leq F(W_t) - \eta\langle\nabla F(W_t), \mathbb{E}_t[\tilde{\mathbf{g}}_t]\rangle + \frac{L\eta^2}{2}\mathbb{E}_t[\|\tilde{\mathbf{g}}_t\|^2]. \tag{24}$$

We handle the inner product term $\langle \nabla F(W_t), \mathbb{E}_t[\tilde{\mathbf{g}}_t] \rangle$. To relate it to the norm of the global gradient $\nabla F(W_t)$:

$$\begin{aligned}
\langle \nabla F(W_t), \mathbb{E}_t[\tilde{\mathbf{g}}_t] \rangle &= \langle \nabla F(W_t), \nabla F(W_t) \rangle + \langle \nabla F(W_t), \mathbb{E}_t[\tilde{\mathbf{g}}_t] - \nabla F(W_t) \rangle \\
&= \|\nabla F(W_t)\|^2 + \langle \nabla F(W_t), b_t \rangle.
\end{aligned} \tag{25}$$

Here, we introduce the symbol $b_t = \mathbb{E}[\tilde{\mathbf{g}}_t] - \nabla F(W_t)$ to represent our strategy's bias toward suppressing conflicting rows.

Substitute the above expression back into the expected inequality:

$$\mathbb{E}_t[F(W_{t+1})] \le F(W_t) - \eta \|\nabla F(W_t)\|^2 - \eta \langle \nabla F(W_t), b_t \rangle + \frac{L\eta^2}{2} \mathbb{E}_t[\|\tilde{\mathbf{g}}_t\|^2]. \tag{26}$$

According to Assumption 4, this deviation is bounded $|\mathbf{b}_t| \le \xi$. Then we utilize the Cauchy-Schwarz Inequality and Young's Inequality to transform the bias term $-\eta \langle \nabla F(W_t), \mathbf{b}_t \rangle$ into a more manageable form:

$$\begin{aligned}
-\eta \langle \nabla F(W_t), b_t \rangle &\le \eta |\langle \nabla F(W_t), b_t \rangle| \\
&\le \eta \|\nabla F(W_t)\| \cdot \|b_t\| \\
&\le \eta \left( \frac{1}{2} \|\nabla F(W_t)\|^2 + \frac{1}{2} \|b_t\|^2 \right) \\
&\le \frac{\eta}{2} \|\nabla F(W_t)\|^2 + \frac{\eta}{2} \xi^2.
\end{aligned} \tag{27}$$

Substitute this upper bound back into the original expression:

$$\begin{aligned}
\mathbb{E}_t[F(W_{t+1})] &\le F(W_t) - \eta \|\nabla F(W_t)\|^2 + \eta \left( \frac{1}{2} \|\nabla F(W_t)\|^2 + \frac{1}{2} \xi^2 \right) + \frac{L\eta^2}{2} \mathbb{E}_t[\|\tilde{g}_t\|^2] \\
&= F(W_t) - \frac{\eta}{2} \|\nabla F(W_t)\|^2 + \frac{\eta}{2} \xi^2 + \frac{L\eta^2}{2} \mathbb{E}_t[\|\tilde{g}_t\|^2].
\end{aligned} \tag{28}$$

Based on the conclusion of Theorem 4.1 and Assumptions 2 and 3, the variance of the global update after aggregation satisfies:

$$\begin{aligned}
\mathbb{E}[\|\tilde{g}_t\|^2] &= \mathbb{E}[\|\tilde{\mathbf{g}}_t - \mathbb{E}[\tilde{\mathbf{g}}_t]\|^2] + \|\mathbb{E}[\tilde{\mathbf{g}}_t]\|^2 \\
&\le \frac{1}{K} \left( \sigma_l^2 + \gamma \cdot \sigma_g^2 \right) + G^2.
\end{aligned} \tag{29}$$

Substitute this expression into the inequality:

$$\mathbb{E}_t[F(W_{t+1})] \le F(W_t) - \frac{\eta}{2} \|\nabla F(W_t)\|^2 + \frac{\eta}{2} \xi^2 + \frac{L\eta^2}{2} \left( \frac{\sigma_l^2 + \gamma \cdot \sigma_g^2}{K} + G^2 \right). \tag{30}$$

Sum the above inequalities from $t = 0$ to $T - 1$ and rearrange the terms:

$$\frac{1}{T} \sum_{t=0}^{T-1} \mathbb{E}[\|\nabla F(W_t)\|^2] \le \frac{2(F(W_0) - F^*)}{\eta T} + \xi^2 + L\eta \left( \frac{\sigma_l^2 + \gamma \cdot \sigma_g^2}{K} + G^2 \right), \tag{31}$$

where $F^*$ is a lower bound on the loss function.

Select the learning rate $\eta$ as $\eta = \frac{1}{\sqrt{KT}}$ and substitute it into the above equation:

$$\frac{1}{T} \sum_{t=0}^{T-1} \mathbb{E}[\|\nabla F(W_t)\|^2] \le \frac{2(F(W_0) - F^*)}{\sqrt{KT}} + \xi^2 + \frac{L}{\sqrt{KT}} \left( \frac{\sigma_l^2 + \gamma \cdot \sigma_g^2}{K} + G^2 \right). \tag{32}$$

Neglecting the constant term and the diminishing term, we obtain Theorem 4.2:

$$\frac{1}{T} \sum_{t=1}^{T} \mathbb{E}[\|\nabla \mathcal{F}(W_t)\|^2] \le \mathcal{O}(\frac{1}{\sqrt{KT}}) + \mathcal{O}(\frac{\sigma_l^2 + \gamma \cdot \sigma_g^2}{K}) + \mathcal{O}(\xi). \tag{33}$$

## B CALCULATION OF CROSS-TERM NOISE

To simplify the calculation process, we derive Eq.3 from Eq.4.

Using the distributive property, we can expand the expression:

$$Ideal - Naive = \sum_{k=1}^{K} p_k(B_k - \bar{B})(A_k - \bar{A})$$

$$= \sum_{k=1}^{K} p_k(B_kA_k - B_k\bar{A} - \bar{B}A_k + \bar{B}\bar{A}) \tag{34}$$

$$= \sum_{k=1}^{K} p_kB_kA_k - \sum_{k=1}^{K} p_kB_k\bar{A} - \sum_{k=1}^{K} p_k\bar{B}A_k + \sum_{k=1}^{K} p_k\bar{B}\bar{A}$$

The second term $\bar{B} = \sum_{k=0}^{K} p_kB_k$ is a constant quantity, transformed as follows:

$$\sum_{k=1}^{K} p_kB_k\bar{A} = (\sum_{k=1}^{K} p_kB_k)\bar{A} = \bar{B}\bar{A} \tag{35}$$

Similarly,

$$\sum_{k=1}^{K} p_k\bar{B}A_k = \bar{B}(\sum_{k=1}^{K} p_kA_k) = \bar{B}\bar{A} \tag{36}$$

The last term is:

$$\sum_{k=1}^{K} p_k\bar{B}\bar{A} = (\sum_{k=1}^{K} p_k)\bar{B}\bar{A} = \bar{B}\bar{A}, \;\; where \sum_{k=1}^{K} p_k = 1 \tag{37}$$

Substituting these into Eq.34 yields:

$$Ideal - Naive = \sum_{k=1}^{K} p_kB_kA_k - \bar{B}\bar{A}$$

$$= \sum_{k=1}^{K} p_kB_kA_k - \sum_{k=1}^{K} p_kB_k \cdot \sum_{k=1}^{K} p_kA_k \tag{38}$$

$$= \underbrace{[W_0 + (p_1B_1A_1 + ... + p_KB_KA_K)]}_{Ideal\ aggregation} -$$

$$\underbrace{[W_0 + (p_1B_1 + ... + p_KB_K)(p_1A_1 + ... + p_KA_K)]}_{Naive\ aggregation}$$

## C ADDITIONAL EXPERIMENTAL DETAILS

### C.1 DETECTION OF CONFLICT VALUES AFTER LORA REGULATION

To demonstrate the effectiveness of our fine-grained conflict joint regulation mechanism, we visualized the average conflict values of LoRA during the training process of the LLAMA model. To ensure fairness, we calculated the conflict values before the regulation mechanism was applied in each round of aggregation, which better reflects the efficacy of our method. Fig.4 illustrates the changes in conflict values for 60 layers of LoRA in the LLAMA model. The results indicate a decline in conflict values across layers after our regulation, suggesting that the mechanism effectively addresses the fine-grained row conflict issues within LoRA. Additionally, the theoretical analysis section (Theorem 4.1 A.1) provides corresponding validation for these findings.

Furthermore, we compare the parameter space of the new round low-rank matrices obtained through singular value decomposition with our method during the training process of the LLaMA model, as

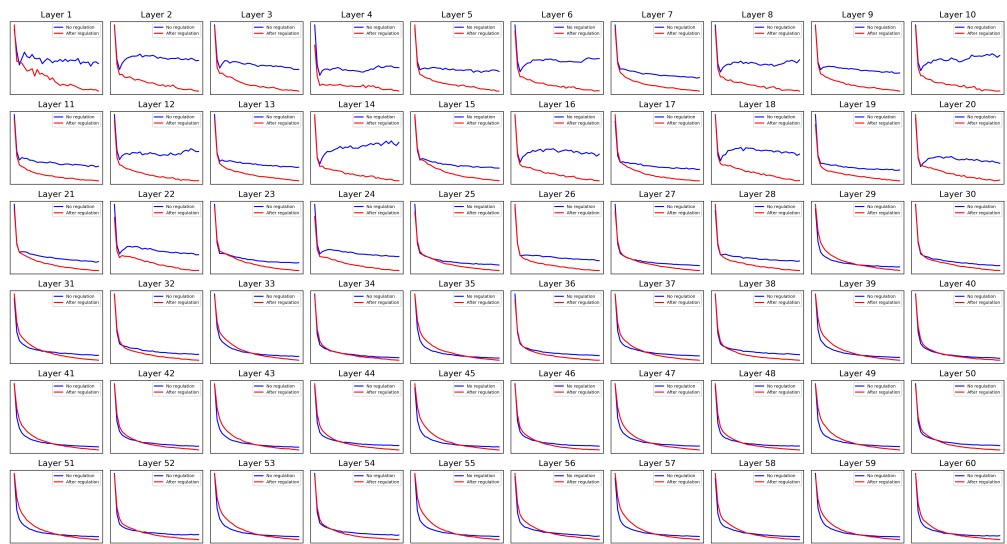

Figure 4: Visualization of the average conflict value across layers of regulated LoRA as the number of rounds changes.

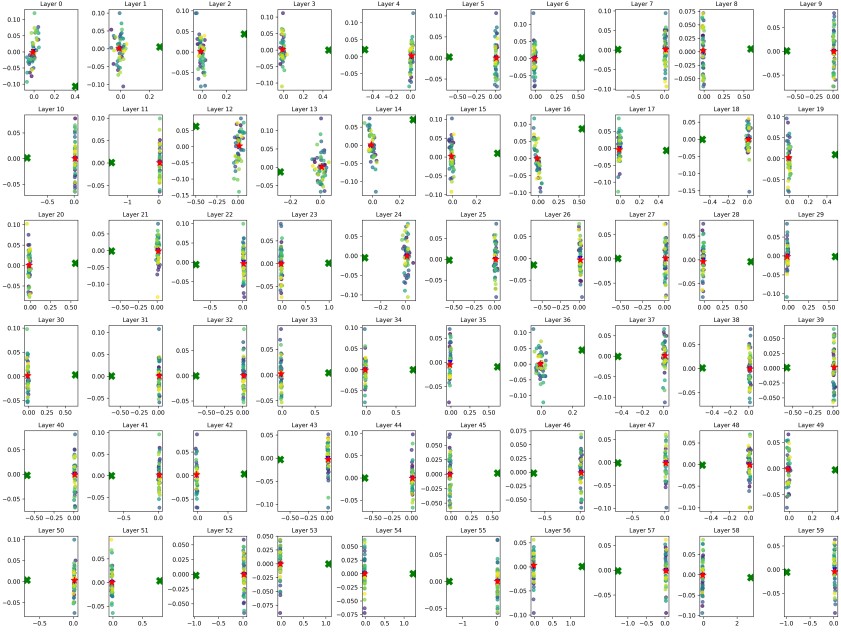

Figure 5: Visualization of low-rank matrix $B$ decomposition comparison in LLAMA model.

illustrated in Fig.5 and Fig.6. The results indicate that the parameter spaces of the low-rank matrices B and A, after singular value decomposition, are significantly distant from the parameter spaces among clients, which is detrimental to training consistency. Our strategy, by introducing a regularization term, ensures that the parameters of the new round remain within the original parameter space and effectively reduces cross-term noise, thereby smoothing the training process in federated learning.

## C.2 DATASETS

We conduct natural language generation and visual tasks on three datasets as follows:

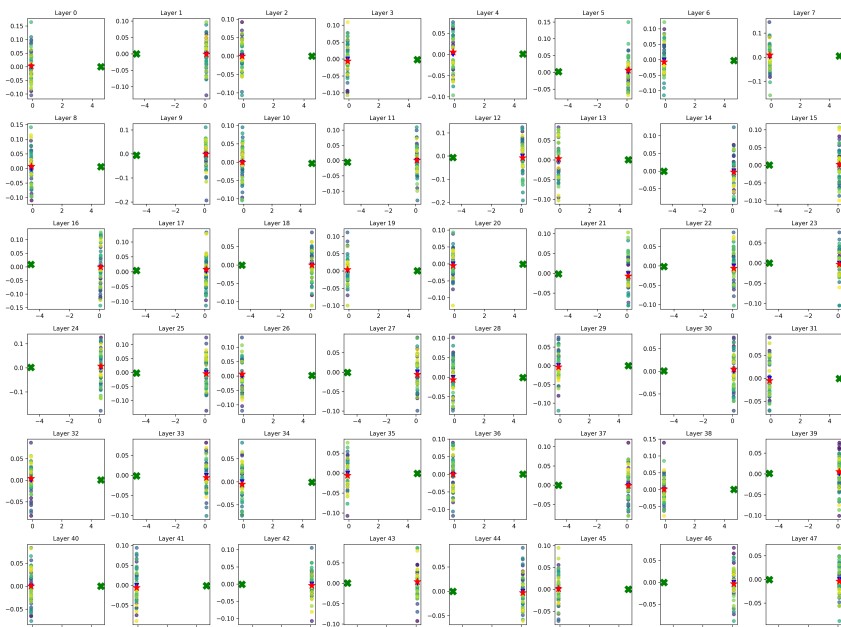

Figure 6: Visualization of low-rank matrix $A$ decomposition comparison in LLAMA model.

- **Alpaca-GPT4 (Peng et al. (2023))**: This dataset contains general knowledge responses generated by GPT-4 using Alpaca prompts, intended for fine-tuning Foundation Models (FMs). It comprises a collection of approximately 52K response samples.

- **OpenR1-Math**: This is a large-scale dataset for mathematical reasoning, comprising 220,000 math problems with correct answers generated by DeepSeek R1. We conduct federated experiments using the default 94K dataset.

- **Places365 (Zhou et al. (2017))**: The Places dataset comprises over 10 million images across more than 400 distinct scene categories, with each category containing between 5,000 and 30,000 training images. We employ a subset of this dataset for the 365-class scene recognition task, totaling approximately 1.8 million images.

## C.3 BENCHMARKS

We evaluate fine-tuned FMs on the following test benchmarks:

- **MMLU (Hendrycks et al. (2021))**: This is a large-scale, multi-task assessment comprising multiple-choice questions from diverse fields of knowledge. It includes 57 tasks covering elementary mathematics, American history, computer science, law, and more.

- **Vicuna (Chiang et al. (2023))**: This benchmark test comprises 80 general knowledge questions. As an open-ended assessment, scoring is based on the model's responses to the questions, evaluated using models such as GPT.

- **MT-Bench (Zheng et al. (2023))**: This benchmark comprises 80 high-quality multi-turn dialogue questions designed to evaluate multi-turn conversation and instruction-following capabilities. During dataset construction, eight common user prompt categories are identified: writing, role-playing, information extraction, reasoning, mathematics, coding, STEM, and social sciences. Each question includes two dialogue segments and is used as an open-ended test.

- **Math500 (Lightman et al. (2023))**: MATH-500 Benchmark Test, developed by OpenAI, is designed to evaluate the mathematical capabilities of its latest models. The test comprises 500 challenging mathematical competition problems aimed at pushing the models to their limits, assessing their reasoning and problem-solving abilities on complex mathematical issues across fields such as algebra, geometry, and probability.

- **Gpqa (Rein et al. (2024))**: GPQA is a question-answering benchmark dataset specifically designed for graduate-level students. This dataset comprises a series of carefully crafted question-answer pairs spanning diverse knowledge domains, including biology, physics, chemistry, and other disciplines. We evaluate only the mathematical questions within this dataset.
- **GSM8K (Cobbe et al. (2021))**: GSM8K (Grade School Math 8K) is a dataset comprising 8.5K high-quality, linguistically diverse elementary math word problems. This dataset supports question-answering tasks involving fundamental mathematical problems requiring multi-step reasoning. As an evaluation set, we utilize 1.32K test cases from this dataset.

## D LLMs USAGE STATEMENT

In accordance with relevant policy requirements, we hereby provide a detailed explanation regarding large language models. During the preparation of this paper, LLMs are utilized to assist in translating English texts and checking for English grammatical errors. Beyond these purposes, no other functionalities of large language models are employed.

