# OpenReview forum: "Rethinking LoRA Aggregation for Federated Fine-tuning of Foundation Models"
_ICLR.cc/2026/Conference — Submitted to ICLR 2026_

### Official Review · Reviewer_1dtz · 2025-10-28

**Soundness:** 3
**Presentation:** 3
**Contribution:** 3
**Rating:** 6
**Confidence:** 4

**Summary:**

This work finds that LoRA parameter aggregation among clients gives rise to fine-grained conflicts and introduces the cross-term
noise interference for subsequent rounds. Based on these findings, the authors propose a Harmonious Federated Low-Rank Adaption method (HFLoRA), which first detects conflicts in LoRA row update directions between clients through a fine-grained joint regulation mechanism, then imposes inhibitory constraints on anomalous conflict rows using scaling factors. In addition, they design a global LoRA consistent re-decomposition strategy that further mitigates the impact of cross-term noise on FL by computing a pair of optimal low-rank matrices from the aggregated noise-free global LoRA.

**Strengths:**

1. The motivation of this work is clear.
2. The authors investigate the parameter conflicts across clients and cross-term noise issues in LoRA aggregation within federated learning, offering insights into the causes of performance degradation.
3. The proposed HFLoRA mitigates LoRA row aggregation conflicts and implements the global LoRA consistent re-decomposition
to enhance FL performance under data heterogeneity.
4. Extensive experiments on NLP and CV tasks demonstrate the superiority of HFLoRA over existing methods.

**Weaknesses:**

1. The font in Figure 1(a) is a bit small, and the experimental settings are not clearly explained. Is this the result for the IID case or the non-IID case? Will data heterogeneity have an impact on this?

**Questions:**

Though this work is well-motivated and well-written, I have some questions about it:
1. Regarding the statements in lines 261-264. Is there any error in using the average of all clients as the anchor point? After all, this average value does not represent the consensus update direction. If this average can be used as the anchor point, does it imply that it is regarded as the consensus update direction? If so, why not simply use the average?
2. In lines 279-280: "Influenced by the experimental results (Fig.1a), we set δ as the percentile 85% of the statistical distribution of the conflict values C." Will the different models and datasets here be different? Is it optimal to set it directly to 85%?
3. In lines 280-282: "This threshold will be adjusted in accordance with the changes in parameter distribution during each round of aggregation, accommodating the dynamic instability and convergence of the model training at different stages." How exactly is this threshold adjusted?
4. How is the optimization problem in Equation (11) solved? What is the time complexity? Furthermore, an experiment needs to be added. In this case, what is the true difference between the learned \\( B^{t+1} \\) and \\( A^{t+1}  \\) and \\( \Delta W_{\text{global}} \\)? Has the problem of aggregation errors really been resolved?
5. In lines 322-323: "The second term acts as a consistent regularization to prevent the optimized \\( B^{t+1} \\) and \\( A^{t+1}  \\) from deviating from the original parameters." Why is this term added? The original average \\( A_i \\) probably doesn't make much sense either, does it? Moreover, an ablation study needs to be included to evaluate the removal of these two terms.

---

### Official Review · Reviewer_bUNw · 2025-10-31

**Soundness:** 3
**Presentation:** 3
**Contribution:** 2
**Rating:** 4
**Confidence:** 5

**Summary:**

The paper investigates why standard LoRA aggregation under data heterogeneity underperforms in federated fine-tuning of foundation models and proposes **HFLoRA**, a two-part remedy.

Firstly, the server forms the LoRA weight matrix per client and calculates a *conflict score* for each row using cosine distance to the mean-normalized row direction across clients. Rows whose conflict exceeds a threshold get scaled down by an exponential decay factor, and the paper proves variance reduction and convergence under standard FL assumptions.

Secondly, instead of aggregating $B$ and $A$ separately, the server first aggregates the regulated full-rank LoRA matrices into a noise‑free $\Delta W_g=\sum_k p_k \Delta \widetilde{W}_k$ and then solves an optimization problem to obtain the next‑round low‑rank factors $(A^{t+1}, B^{t+1})$  that reconstruct $\Delta W_g$ and remain close to the averaged $(A^t, B^t)$ to ensure parameter‑space consistency.

Experiments cover LLaMA‑2‑7B instruction-tuning on Alpaca‑GPT4 and OpenR1‑Math, as well as ViT on Places365. HFLoRA generally improves over other baselines.

---
### LLM usage disclosure (reviewer)
I used GPT‑5 to help polish and organize this review; I take full responsibility for the content.

**Strengths:**

**Clear diagnosis of two concrete aggregation pathologies.** The cross‑term argument is clean and instructive, and the row‑level conflict visualization is compelling and clearly shows that only a minority of rows are strongly conflicting.

**Simple server‑side implementation.** Computing row conflicts, scaling, and then aggregating is conceptually straightforward and compatible with standard communication.

**Theoretical backing** Thm. 4.1–4.2 give variance reduction and a non‑convex FL‑style convergence rate under standard smoothness/variance assumptions plus a consensus update assumption tailored to the row‑level design. While high‑level, this is consistent with contemporary FL analyses.

**Weaknesses:**

1. **Key assumptions and sensitivity not fully explored:** The core Consensus Update assumption posits that client row updates concentrate around a global direction with slight systematic bias. In realistic, highly non‑IID regimes (disjoint tasks, label/feature skew), this may fail. The method might over‑suppress legitimate client‑specific information. A robustness study varying heterogeneity (especially on NLG) and stress‑testing beyond $\beta=0.5,1.0$ would strengthen the claim.

The conflict threshold $\delta$ is set at the 85th percentile. The paper does not demonstrate sensitivity to $\delta$ or $\lambda$, nor does it provide a principled approach to setting them. This matters because $\gamma_j$ can effectively zero out rows.

2. **Details of the re‑decomposition solver are missing.**
   Equation (11) defines a bi-linear least-squares problem with regularizers; however, the paper does not specify the optimization routine or configuration.

3. **Heterogeneous ranks: how is the broadcast handled?** The method claims to support $r_k$ heterogeneity across clients, but it is unclear what rank the server uses for $(A^{t+1},B^{t+1})$ and how clients with different $r_k$ ingest the broadcast. Do clients project to their local ranks (and if so, how)? More detail is needed on the global rank selection and compatibility across rounds.

4. **Computation and memory cost at the server**: This method requires reconstruction and operation on the full-rank matrices at the server, which increases the memory and computation significantly in comparison to low-rank aggregation.
The cost of the server for both reconstruction and decomposition is not reported, and the scalability is not clear.

**Questions:**

Please look at the weaknesses.

---

### Official Review · Reviewer_BJ4r · 2025-10-31

**Soundness:** 2
**Presentation:** 2
**Contribution:** 2
**Rating:** 2
**Confidence:** 5

**Summary:**

This paper investigates the LoRA row conflict and cross-term noise problems in federated fine-tuning of foundation models. To address these issues, the proposed HFLoRA method combines fine-grained conflict regulation with global LoRA re-decomposition, achieving superior performance across diverse LoRA configurations.

**Strengths:**

The proposed method offers a well-motivated and effective solution to key challenges in federated fine-tuning with low-rank adaptations.

**Weaknesses:**

1. The experimental setup in Fig. 1(a) lacks sufficient explanation. What is the specific LoRA configuration used? Under which data setting were these experiments conducted? Additionally, which training round does this figure represent, early or late in the training process?

2. There is a conceptual error in the discussion of FedSA (share-A)-LoRA in Line 156. In this setting, only matrix A is shared across clients, not matrix B.

3. As a follow-up, since FedSA-LoRA shares only matrix A, and matrix B is typically zero-initialized, what global model was used in your evaluations? Based on their project’s description, it appears to be more aligned with a personalized FL method. Could you clarify?

4. The issue of cross-term noise is not novel and has been discussed in many prior work.

5. The paper omits relevant prior work on layer-wise efficient federated fine-tuning methods, such as FedRA [1] and Fed-HeLLo [2].

6. Could you provide visualizations of the cosine distance for different tasks (visual v.s. language) as referenced in Line 259? These would help illustrate the variation in optimal directions across tasks. Are there any insights or takeaways from such visualizations?

7. The threshold set in Line 279 appears to be empirically chosen. Was it evaluated under different tasks or heterogeneity settings?

8. How is the optimization problem in Equation (11) solved? The paper does not provide sufficient details on the solution procedure or algorithm.

9. There are several minor typos and formatting issues:

– Line 83: use “this issue” twice

– Line 245: “LORA” should follow consistent capitalization

– Line 257: comma appears incorrectly at the beginning of the line

10. Have you observed any similar conflicts in the columns (or channels, if applicable) of the weight matrices? If so, further analysis would be helpful.

[1] FedRA: A Random Allocation Strategy for Federated Tuning to Unleash the Power of Heterogeneous Clients. ECCV 2024.

[2] Fed-HeLLo: Efficient Federated Foundation Model Fine-Tuning with Heterogeneous LoRA Allocation. IEEE TNNLS 2025.

**Questions:**

See above.

---

### Official Review · Reviewer_W2Sx · 2025-11-01

**Soundness:** 3
**Presentation:** 2
**Contribution:** 2
**Rating:** 4
**Confidence:** 3

**Summary:**

The paper studies parameter-efficient federated fine-tuning with LoRA adapters and argues that two aggregation pathologies hinder performance under data heterogeneity:
(i) Row-wise conflicts in the LoRA weight ΔW=BA, where a minority of row updates across clients point in incompatible directions; and (ii) cross-term noise from naively averaging A and B separately rather than aggregating the actual update ΔW.

**Strengths:**

- The paper studies why naively averaging LoRA adapters across clients harms federated fine-tuning under non-IID data.
- The paper crisply formalizes “cross‑term noise” in LoRA aggregation via the identity
$∑p_k B_k A_k ≠ (∑p_k B_k)(∑p_k A_k)$
(Eq. 3) and derives a compact expression for their difference (Eq. 4).
- Broad evaluation across NLG and vision

**Weaknesses:**

- The paper argues lower communication than stacking. FLoRA indeed yields a global rank that grows with the number of clients (stacking), which can raise download costs; and discusses this trade-off explicitly. A numeric apples-to-apples communication/runtime study (same number of LoRA injected layers, same K, same ranks) would make Table 3 more convincing.
- Limited analysis against LoRA-FAIR and FRLoRA, which share HFLoRA’s core goals. Both methods explicitly mitigate aggregation bias and initialization drift. More fine-grained comparisons e.g., robustness under varying heterogeneity β, cold-start behavior, and per-round convergence speed, would clarify whether HFLoRA truly outperforms them rather than offering a lateral variant.
- Experiments use K ≤ 30 (vision) and ≤ 20 (LLM). In realistic cross-device FL, K ≫ 100, and many clients participate only intermittently. The per-round server overhead of reconstructing ΔWₖ = BₖAₖ and solving Eq. (11) could become significant. A concrete cost analysis (server FLOPs, GPU time, memory vs. K and r) is missing.
- The method fixes δ to the 85th percentile of C and applies a decay λ (Eq. 9) but provides no study of how these choices affect performance or stability. Since the approach relies on suppressing a small subset of rows, the authors should plot accuracy and variance versus δ (70/80/90/95th percentiles), λ, and τ, as well as the fraction of suppressed rows per round.
- Table 4 claims support for heterogeneous ranks, but Eq. (11) produces a single (A^{t+1}, B^{t+1}). How are clients with different rₖ handled via per-client SVD truncation, shared r projection, or multiple decompositions? What is the additional server cost? Baselines like FLoRA and FlexLoRA explicitly describe these redistribution strategies.

**Questions:**

- Can you provide a clearer trade-off analysis among FLoRA, LoRA-FAIR, and FRLoRA, covering accuracy, communication, runtime, and heterogeneity robustness?
- What optimization algorithm is used to solve Eq. (11) e.g., alternating least squares, proximal gradient, or an SVD-based closed form? how many iterations per round, and what are the typical time/memory costs per layer for LLaMA 2 7B? Does the added computation offset the claimed communication savings?
- When clients use heterogeneous ranks, how is the server’s rank rt+1 chosen per layer? Is it fixed or adaptive? Any risk of rank deficiency hurting downstream performance?
- Can you report the time required per round to transmit load and re-initialize LoRA weights on clients, and compare HFLoRA’s total per-round runtime (aggregation + re-decomposition + distribution) with baseline methods?

---

### Meta-Review · Area_Chair_8QoJ · 2026-01-05

**Summary:**

The reviewers were generally on the negative side. The key concerns raised were as follows.
- The description of the proposed framework seems unclear---reviewers have noted multiple instances of technical components which not fully explained, or analyses that lack a concrete description.
- Justification of the proposed method is insufficient.
- Shortage of analyses and ablations.

Unfortunately, the authors did not respond to the reviews during the discussion period. Thus the concerns remain essentially unchallenged. I have also checked the paper, and agree with the reviewers comments in general.

Hence, I recommend rejection.

**Reviewer Concerns:**

As the authors did not respond, all concerns are still outstanding.

**Reviewer Scores:**

None of the reviewers may have raised the score.

---

### Decision · Program_Chairs · 2026-01-26

Reject